# A Parallel Compression Pipeline for Improving GPU Virtualization Data Transfers

**DOI:** 10.3390/s24144649

**Published:** 2024-07-17

**Authors:** Cristian Peñaranda, Carlos Reaño, Federico Silla

**Affiliations:** 1Departamento de Informática de Sistemas y Computadores, Universitat Politècnica de València, 46022 Valencia, Spain; 2Departament d’Informàtica, Escola Tècnica Superior d’Enginyeria (ETSE-UV), Universitat de València, 46010 Valencia, Spain

**Keywords:** deep learning, on-the-fly compression, parallel compression pipeline, network bandwidth, remote GPU virtualization, CUDA, rCUDA

## Abstract

GPUs are commonly used to accelerate the execution of applications in domains such as deep learning. Deep learning applications are applied to an increasing variety of scenarios, with edge computing being one of them. However, edge devices present severe computing power and energy limitations. In this context, the use of remote GPU virtualization solutions is an efficient way to address these concerns. Nevertheless, the limited network bandwidth might be an issue. This limitation can be alleviated by leveraging on-the-fly compression within the communication layer of remote GPU virtualization solutions. In this way, data exchanged with the remote GPU is transparently compressed before being transmitted, thus increasing network bandwidth in practice. In this paper, we present the implementation of a parallel compression pipeline designed to be used within remote GPU virtualization solutions. A thorough performance analysis shows that network bandwidth can be increased by a factor of up to 2×.

## 1. Introduction

Artificial intelligence (AI) is progressively becoming part of our daily life thanks to the use of deep learning models based on neural networks. These networks are not new. However, their intensive computing requirements have been recently addressed by leveraging accelerators such as Graphics Processing Units (GPUs).

Typically, deep learning applications are tied to using powerful computers equipped with last-generation GPUs [1]. Nevertheless, although this execution environment is fine for many application domains, there are other novel domains where using lighter devices could be more advisable. These novel domains include, for instance, environments where large amounts of data are captured in IoT (Internet of Things) devices to perform timely and high-quality decisions supported by deep learning models [2].

In these novel application domains, data captured by IoT devices could be sent to a powerful GPU-equipped computer in the cloud, where the deep learning model is executed, and decisions would be sent back to the IoT device. However, other approaches are also feasible. For example, edge/fog computing [3] leverages the decentralization scheme by carrying out computations close to (or even at) the IoT element that captures the data. Processing captured data close to the location of the IoT device sampling the real world provides energy savings and scalability, among other benefits.

However, IoT devices present severe power limitations, limiting their computing capabilities [4]. In this regard, edge devices used in IoT typically rely on ultra-low power solutions such as ARM-based CPUs, which are unsuitable scenarios for executing deep learning models. Even if these ARM-based devices include an accelerator (GPU), their total computing power is quite small compared to that of a last-generation GPU.

An alternative approach to increase the computing capabilities of edge devices is the use of remote virtualization of accelerators [5]. This mechanism transparently provides GPUs located at remote nodes to applications being executed in the edge device. Thus, edge devices do not pay for an energy increase, and simultaneously they can access a large number of powerful remote GPUs. As remote GPU virtualization provides GPUs transparently to applications, their source code does not need to be modified.

Notice that accessing remote GPUs requires a network bandwidth typically larger than the one available in IoT deployments. In this way, the small bandwidth available in IoT could limit the applicability of remote GPU mechanisms in this context. However, it is possible to artificially increase network bandwidth by using compression techniques. That is, data exchanged with the remote GPU could be compressed on-the-fly before being injected into the network, and once they arrive at the remote GPU server, they would be decompressed before being copied to the GPU. On the way back, data transmitted by the remote GPU server would also undergo this process. It is important to remark that this on-the-fly compression is carried out by the communication layer of the remote GPU virtualization solution, and thus, it is transparent to the application, which is unaware of its data being compressed. In summary, if the compression and decompression stages are properly designed, it could be possible that they provide the benefit of reducing the size of data exchanged with the remote GPU (thus increasing network bandwidth in practice), whereas no extra latency is added to transmissions.

This paper presents the design and implementation of a parallel compression pipeline system for remote GPU virtualization solutions. We also perform a thorough performance analysis to investigate how the different parameters of the pipeline influence the overall bandwidth attained by the system. It should be noted that we do not propose novel compression algorithms in this paper. On the contrary, we use well-known compression algorithms to implement and evaluate our parallel pipeline proposal. The key contributions of this work are as follows:We design and implement a parallel compression pipeline system for remote GPU virtualization, addressing constraints such as low-performance networks (commonly present in IoT environments) or data types (lineal, random, sparse, and traced data).We conduct a comprehensive performance analysis to analyze the impact of the different parameters on the implemented solution, providing insights for optimizing such systems.We demonstrate how existing compression libraries can enhance network bandwidth using parallel pipeline mechanisms without introducing significant latency.

The rest of the paper is organized as follows. Section 2 presents the background required to better understand the research presented in this paper. Next, in Section 3, our proposal is introduced and later evaluated in Section 4. Finally, conclusions and future work are presented in Section 5.

## 2. Related Work

To better understand the rest of the paper, this section presents some background on remote GPU virtualization systems (Section 2.1) and compression solutions (Section 2.2).

### 2.1. Remote GPU Virtualization Systems

Remote GPU virtualization solutions allow a GPU application to be executed on a computer that does not own a GPU. To that end, these solutions use a GPU located in another computer. In the area of machine learning, NVIDIA GPUs have had a huge impact. For that reason, in this paper, we focus on systems supporting the CUDA technology developed by NVIDIA [6].

Remote GPU virtualization solutions typically follow a client–server architecture, as  Figure 1 illustrates. A library provides the same API as CUDA on the client side, where the GPU application is running. That allows the system to intercept all GPU requests made by the application. This library processes the intercepted GPU requests and sends them to the server side over the network. A daemon is listening for new GPU requests on the server side, where the GPU is physically installed. Once the daemon receives a GPU request, it executes the request on the GPU and sends the result back to the client side. It should be noted that application developers do not have to modify the application source code in order to use remote GPU virtualization solutions. The GPU application is not aware that the request has been processed by a virtual GPU instead of a local real one. Using remote GPUs is transparent to the application because all the required details are managed by the remote GPU virtualization solution.

There are different remote GPU virtualization solutions. GVirtuS [7] and rCUDA [8] are probably the most popular ones. Both of them present a similar client–server architecture, although rCUDA supports more CUDA features. For that reason, we use the rCUDA middleware in this work.

### 2.2. Compression Solutions

The volume of data used increases day by day [9]. Compression algorithms are an excellent approach to reducing the volume of data stored or sent over the network, for instance. In this sense, different areas have used compression libraries in the past to achieve performance improvements.

Concerning inter-process communications, Vega et al. [10] use delta compression techniques in fusion databases to save storage and increase the bandwidth. In contrast, Hansson and Karlsson [11] utilize the well-known compression libraries Lzo, Lzfx, Lzw, Lzma, Bzip2, and Lz4 to support lossless message compression. Both studies obtain promising results, and the last one concludes that Lz4, Lzo, and Lzfx are the fastest libraries.

Liang and Li [12] and Uthayakumar et al. [13] focus on wireless sensor networks. Due to the limitations of these devices on energy, bandwidth, memory, and processing capabilities, Liang and Li [12] compare a new proposal with popular lossless compression algorithms analyzing power and robustness. Alternatively, Uthayakumar et al. [13] introduce a highly reliable and low-complexity compression scheme using a neighborhood correlation sequence algorithm.

There are also some experiments focused on improving network bandwidth. Welton et al. [14] investigate compression services that compress and decompress data before transferring it over the network. Their exploration uses Bzip2, Zlib, and Lzo. Results show the relevance of the data used to evaluate compression services. Also, Wiseman [15] proposes a transmission system that is able to select the intensity of the compression algorithm depending on the CPU cycles and the bandwidth available. On the other hand, Routray et al. [16] enumerate different compression algorithms used in Internet of Things environments over low bandwidth networks, such as Narrowband Internet of Things and Long-Term Evolution Machine Type Communication. They argue in favor of using compression systems to reduce resources and data transfers done in these networks, but they do not provide any experimental results.

Hu [17] studies the possibility of compressing the data sent over the network based on the available bandwidth in a 10 Mbps network. The author proposes to create a simple pipeline that splits the data into different chunks and, depending on the available network bandwidth, it calculates if a chunk must be compressed or not before sending it. Although the different pipeline stages do not overlap, the results are promising.

Krintz and Sucu [18] present an adaptive on-the-fly compression system to transmit data over 100 Mbps and 1.7 Mbps networks. The system splits the data into chunks of 32 KB, checks the available bandwidth on the network, and determines whether it compresses the chunk (using Lzo, Zlib, and Bzip2) before sending it. Although data are split into different chunks, unfortunately, the stages of the system do not overlap.

Peterson and Reiher [19] present a system that decides whether to compress or not a specific chunk of data using Lzo, Zlib, Bzip2, and Xz compression libraries. Data are split into 32 KB chunks, and a parallel pipeline compression system is used, hiding the compression and decompression time behind the transfer time. However, despite the acceptable performance, the study does not compare the results obtained with a compression system without a pipeline nor does it evaluate different pipeline configurations.

Chowdhury et al. [20] create a novel two-step compression scheme that exploits temporal correlation in the individual streams to be transmitted in order to increase bandwidth savings. This study obtains a significant reduction in the network resources required at the expense of being a lossy solution.

Kim et al. [21] introduce a selective data-compression scheme based on data prediction. In this study, they apply LZ77 encoding and dynamic Huffman coding to compress data. The prediction method reduces wasted computing resources by avoiding coding data when necessary. Using this selective compression, they can choose a dynamic or static Huffman codification to code the result of the LZ77 reduction. The authors also show a compression pipeline in which LZ77 encoding and Huffman coding overlap. Although the study focuses on using Huffman codification, there is no research on LZ77 compression. Both algorithms should be considered; otherwise, the resulting data might be larger than the input data, and computing resources would have been wasted.

Finally, Penaranda et al. [22] present the Smash compression abstraction library. It contains 41 different compression libraries, which can be used with different configurations.

Several of the previously mentioned studies focus on enhancing network communication. However, none of these studies provide tools or detailed information on how to implement them. In this article, we develop a parallel mechanism to improve communications in remote GPU virtualization systems. The publication that most closely aligns with our work is the system created by Peterson and Reiher [19]. However, this research lacks prior studies regarding the configuration utilized, which is always the same regardless of the network employed. We consider that this study is essential to advance the research in the field.

## 3. A Parallel Compression Pipeline for GPU Virtualization Data Transfers

As shown in previous work [23], applying a naive on-the-fly compression approach in the communication layer of remote GPU virtualization solutions such as rCUDA could translate into a good improvement in performance when network bandwidth is relatively low. This naive approach compresses all the data before sending it. Once the compressed data are received, they are decompressed. There is no overlap among compression, transmission, and decompression tasks. Moreover, data are not split into smaller chunks in order to compress, send or decompress them.

Given the promising results of such a naive approach, in this paper, we propose a much more sophisticated solution to further increase performance. More specifically, we propose to research the use of a parallel compression pipeline where compression and decompression stages overlap with data transfers, thus hiding the cost of compressing and decompressing data. Additionally, data to be sent are split into chunks, which make progress along the pipeline. As shown in  Figure 2, applying the well-known pipeline technique could significantly improve performance for two reasons: (1) the total amount of transmitted data is reduced (as with the naive approach) and (2) the compression and the decompression stages overlap with the transmission, thus hiding the latency of these stages.

Figure 3 presents a high-level implementation of the proposed parallel compression pipeline. As can be observed, in the proposal shown in the figure, there are *n* threads that work on *m* data chunks. On the client side, *n* threads work in parallel in order to compress data so that the main thread, which sends compressed data over the network, always has a new compressed data chunk ready to be sent. On the server side, *n* threads work in parallel in order to decompress data chunks as they arrive. In this simple example, a compression thread takes, for instance, 2t to compress a chunk, while the main thread takes, for instance, 1.5t to transfer the compressed chunk. If only one compression thread were used instead of *n*, the main thread would have to wait for 0.5t between the current data transfer and the following one. Having several decompression threads is also necessary to improve performance. Similar to what happens with compression threads, the decompression thread takes 2.3t to decompress the compressed chunk. If only one decompression thread were used instead of *n*, a latency of 0.8t would be added to the decompression time.

In the example in  Figure 3, we can see the importance of appropriately optimizing the different stages of the pipeline. In particular, in order to obtain the most efficient possible pipeline, there are some parameters to consider:Number of compression/decompression threads. Depending on the size of the data to be compressed, compression and decompression times are different. For that reason, it is essential that the main thread, which sends compressed chunks over the network, has the next chunk compressed and ready to be sent before finishing the current chunk transfer. To couple the different speeds of producers and consumer threads, it is necessary to choose the appropriate number of compression/decompression threads.Number of data chunks. Splitting the data into multiple data chunks allows compression/decompression threads not to idle. An adequate number of data chunks should be selected so that those threads always have work to do.Size of data chunks. Choosing the best data chunk size is also important. Compression takes longer with a large data chunk than with a smaller one. However, the compression rate is usually better for a large data chunk than for a smaller one. In addition, better compression could also lead to a faster transfer.

Of course, the ideal scenario is one that provides the pipeline with all the data chunks and all the threads it needs. However, in the real world, that is not always possible due to resource limitations. It is, therefore, necessary to find a trade-off between the resources used and the performance achieved. In the next section, we analyze the influence on the performance of the number of threads, the number of data chunks, and the size of the data chunks.

## 4. Experimental Results

In this section, we present and analyze the experimental results. First, we describe the experimental setup in Section 4.1. In Section 4.2, we present the compression libraries used in the experiments. Next, we introduce the data used in the experiments in Section 4.3. Then, in Section 4.4, we describe the mechanism for choosing the best parameters in the parallel compression pipeline system. Once the best parameters have been chosen, in Section 4.5, we analyze the impact on transfer time of transferring the compressed data using the parallel compression pipeline system. Finally, in Section 4.6, we study an additional improvement consisting of sending or not the compressed chunk depending on whether the compressed chunk is larger or smaller than the uncompressed chunk.

### 4.1. Experimental Setup

In our study, we consider the scenarios shown in  Figure 4 to explore the benefits of using a parallel compression pipeline system:Scenario A represents the initial scenario where the remote GPU virtualization solution rCUDA is used. No compression is used.Scenario B shows an improvement over the previous scenario: naive compression is used to compress data transfers carried out within rCUDA.Scenario C makes use of our parallel compression pipeline system. The compression and decompression stages overlap with data transfers done within rCUDA.

Notice that the compression and decompression stages shown in scenarios B and C are implemented inside the communication layer of rCUDA, and therefore, their use is transparent to the rest of the system. On the other hand, scenarios B and C will use the compression libraries described in Section 4.2.

In the experiments, we use an Intel(R) Xeon(R) CPU E5-2637 v2 3.50 GHz (Santa Clara, CA, USA) as the client node, while the server node is an AMD EPYC 7282 16-Core Processor (Santa Clara, CA, USA) with an NVIDIA A100 GPU (Santa Clara, CA, USA). The connection between the client node and the server node is a 1 Gbps Ethernet wired network. The Linux traffic shaper (i.e., tc) is used to reduce the network bandwidth to that usually available in edge environments, such as WiFi or cell phone networks. More specifically, we reduce the network bandwidth to 100 Mbps and 10 Mbps, studying these networks and the 1 Gbps case.

### 4.2. Compression Libraries Used in the Experiments

In the experiments, we used the compression libraries provided by the Smash [22] compression abstraction library, which is available at https://github.com/cpenaranda/smash (accessed on 15 July 2024). In particular, after examining all the compression libraries that Smash provides, we selected the following four libraries:Snappy [24]. The Google team developed this compression library based on the LZ77 algorithm to obtain a fast compressor instead of focusing on compression.Gipfeli [25]. The Google team also developed this compression library based on LZ77. Gipfeli obtains better compression ratios than Snappy but increases the computation time.Lz4 [26]. This LZ77-based compression library focuses on fast compression and decompression.Lzo [27]. This compression library is another LZ77 derivative. It sacrifices compression and decompression speed for compression ratio.

The reason for selecting the aforementioned libraries is twofold: (1) they provide the best performance and (2) despite being the fastest libraries, they present different features. For instance, Lzo is much more computationally intense than Gipfeli. While some of these libraries have existed for several years, their prevalence in current research papers underscores their ongoing relevance and academic acceptance [28,29,30,31,32,33]. Our objective in employing these libraries is not primarily to conduct a comparative analysis to determine the best option but rather to leverage their differences in functionality within the context of developing a parallel compression pipeline. This approach allows us to explore broader insights relevant to our study.

### 4.3. Data Used in Experiments

In order to analyze the benefits of using a pipelined on-the-fly compression approach to artificially increase network bandwidth, we used the bandwidth test developed by NVIDIA, which can be obtained when obtaining the CUDA software. Nevertheless, given that the properties of compression algorithms greatly depend on the exact data being compressed, we modified the bandwidth test program in order to consider the following four data types, all of them represented in Figure 5:Lineal data. Data start from 0 and increase one by one up to 255. Once the value 255 is reached, they start again from 0.Random data. Data are composed of random numbers between 0 and 255.Sparse data. Data contain a random set of 0s followed by a random set of numbers between 0 and 255.Traced data. Traces from real obtained from TensorFlow applications [22]. These data are actually the data exchanged among the host memory and the GPU memory during the execution of TensorFlow applications. We sized the data in order to make them compatible with the CUDA bandwidth test benchmark.

On the one hand, random data are the worst scenario because the information does not follow any pattern. We expect very low compression ratios and, thus, bad results in general when using this data type. On the other hand, lineal and sparse data contain repetitive information and follow patterns, which can help compression libraries to obtain good performance. Finally, the previous data types were generated synthetically and, therefore, are useful for obtaining some insights from our study. However, using traced data seems to be more representative of the environment targeted by this study. This is why, in the next sections, we will focus on presenting results for this data type. In the Appendix A at the end of the paper, the results for the other data types can be found.

### 4.4. Finding the Best Parameters for the Parallel Compression Pipeline

As mentioned in Section 3, the performance of the parallel compression pipeline depends on (1) the number of concurrent threads used for compressing and decompressing data, (2) the amount of data chunks that are concurrently being processed within the pipeline, and (3) the size of these chunks. Given that the design space can be extremely large, we will only consider the values shown in Table 1 for each parameter. Thus, we will analyze the performance of the pipeline by varying the number of threads, the number of data chunks, and the size of those data chunks according to the values in the table.

On the other hand, notice that if we do not limit the usage of resources (i.e., memory and CPU usage), the parallel compression pipeline system will usually obtain better performance as the amount of available resources increases. For instance, for the parameters in Table 1, we can expect that using 8 threads provides the best performance. However, it is likely that the improvement in performance is smaller as the number of threads increases. That is, we can expect that performance greatly improves when the amount of threads increases from 1 to 2. However, when the amount of threads increases from 2 to 4, we can expect a lower improvement in performance. Similarly, increasing available resources from 4 to 8 threads may not improve performance noticeably. Thus, there is a trade-off between the number of available resources and the performance obtained.

In a real scenario, we are limited by the available resources and should use as few resources as possible. Therefore, we have developed an equation to model the trade-off between the performance obtained and the number of resources used to obtain that performance. We will evaluate performance relative to the number of resources used.

Equation (Equation 2) represents the trade-off between the achieved bandwidth and the resources a given parameter configuration uses. Note that BW represents the bandwidth, Size is defined as the chunk size times the number of chunks, as seen in Equation (Equation 1), and Th is the number of threads. In Equation (Equation 2), we first normalize the bandwidth for each value by dividing it by the minimum bandwidth obtained. Similarly, we normalize the size and the number of threads. Finally, we weigh the normalized values with the user-defined constants α, β, and δ. The sum of those constants must be 1. If the α value is close to 1, the user gives more weight (i.e., prioritizes) bandwidth. In contrast, the β or δ values are close to 1 when the user is concerned about resource utilization. For this study, we selected the values 0.60, 0.15, and 0.25, for α, β, and δ, respectively. We give more weight to bandwidth (α is the highest value) because the performance of our approach greatly depends on it. In addition, in our experimental setup, computing nodes are more limited in terms of CPU (i.e., the number of threads) than in memory (i.e., the size of the data chunks), so δ is higher than β.
(1)Size=ChunkSize×NumberChunks
(2)$$\begin{array}{c} \mathrm{Trade}-\mathrm{Off}\left(x\right)=\left(\frac{BW_{x}}{BW_{min}}\times \alpha \right)-\left(\frac{Size_{x}}{Size_{min}}\times \beta \right)-\left(\frac{Th_{x}}{Th_{min}}\times \delta \right) \end{array}$$

On the other hand, the design space exploration considering all the combinations of parameters in Table 1 is unaffordably large. Therefore, in order to reduce the designed space exploration, we will first focus on analyzing the impact of the size of data chunks. Then, by selecting the best options for chunk size we will further reduce the design space exploration by analyzing the impact of the number of chunks. Finally, we will use those conclusions to study the impact on the performance of the number of threads. Moreover, we will carry out this analysis considering only messages of 8 MB of data. We have chosen this message size because it is large enough to keep the pipeline working while measuring performance. In the next section, we will show the performance of the entire range of message sizes.

Notice that, due to space limitations, we only show results for the Gipfeli compression library in this section when using the 1 Gbps network and the traced data. The results for the rest of the compression libraries, network bandwidths, and data types are available in the Appendix A at the end of the paper.

In the first step, we researched the best data chunk size depending on the number of threads and the number of data chunks. Figure 6 presents the results obtained by the Gipfeli compression library using a 1 Gbps network. As we can see, in all scenarios, achieved bandwidth increases as more resources are devoted to the pipeline, as expected. However, the trade-off between resources and bandwidth (denoted as “TO” in the plots) worsens as the memory used increases. The reason is that the bandwidth does not improve enough to compensate for the memory increase. The chunk size that achieved the best performance using one thread is 2 KB, regardless of the number of chunks, as shown Figure 6a. Similarly, in Figure 6b, we can observe that for almost any number of chunks, the optimal chunk size when using two threads is 4 KB. The only exception is for 32 chunks, where the best chunk size is 2 KB. Finally, when using four and eight threads, regardless of the number of chunks, the best performance is obtained when using chunk sizes of 4 KB, as shown in Figure 6c,d.

Once data chunk sizes have been investigated, the next parameter we study is the number of threads. Figure 7 presents this analysis, where we have set the bests chunk size according to results in Figure 6 for each of the thread counts (1, 2, 4, and 8). As can be observed, as the amount of chunks increases, the trade-off between resources and bandwidth worsens. The reason is that the total size used increases exponentially. Table 2 summarizes the results showing the best configurations. We selected the configuration that obtains better bandwidth.

Finally, we study the optimal number of threads when chunk size and the number of chunks are set according to Figure 6 and Figure 7. Again, we use our equation to obtain the best trade-off between bandwidth and resources used. In Figure 8, we can observe that the bandwidth increases as the number of threads increases from 1 to 2 and from 2 to 4. In the case of 8 threads, bandwidth is reduced. However, the best trade-off is achieved with 2 threads. After having studied the three parameters, we can conclude that for the Gipfeli using a 1 Gbps network, the best configuration is 2 threads, with 4 chunks where chunks have a size of 4 KB.

Due to space limitations, we have only shown plots for the Gipfeli compression library and 1 Gbps network (plots for the rest of the compression libraries and networks are available in the Appendix A). However, Table 3 summarizes the best configurations obtained for all the compression libraries used in this work. As shown in the table, using 1 Gbps, Gipfeli obtains better bandwidth than others, despite being Lzo, the compression library with the best compression ratio. This is because the compression and decompression time is the bottleneck in an environment with 1 Gbps networks, and Lzo takes more time to compress and decompress than Gipfeli. However, we can see the importance of obtaining a good compression ratio by reducing the network bandwidth. Thus, when using 100 Mbps and 10 Mbps networks, Lzo performs better than the other compression libraries because transfer time compensates for the extra time spent in compression and decompression. It can also be seen that when network bandwidth decreases, Lzo requires fewer resources.

### 4.5. Impact on the Performance of the Parallel Compression Pipeline

In the previous section, we selected the best configuration parameters for our parallel compression pipeline system when 8 MB messages were used. In this section, we evaluate the performance of the scenarios described in Section 4.1 for the entire range of message sizes. Figure 9, Figure 10 and Figure 11 illustrate the results obtained using 1 Gbps, 100 Mbps, and 10 Mbps network speeds, respectively. The figures present the results for five different cases: (i) rCUDA without using compression, shown as ‘No Compression’, (ii) rCUDA using the parallel compression pipeline system with Gipfeli, displayed as ‘Gipfeli Pipeline’, (iii) rCUDA with a parallel compression pipeline system using Lzo, referred to as ‘Lzo Pipeline’, (iv) rCUDA using compression with Gipfeli but without pipeline, shown as ‘Gipfeli Naive’, and finally (v) rCUDA using compression with Lzo but without pipeline, displayed as ‘Lzo Naive’. The different cases are evaluated with datasets of different sizes, always using traced data (refer to the Appendix A for other data types). Note that ‘Gipfeli Pipeline’ and ‘Lzo Pipeline’ use the best configuration obtained in Section 4.4, and these configurations depend on the network bandwidth, as shown in Table 3.

Figure 9 presents the results using a 1 Gbps network. As we can see, ‘Gipfeli Pipeline’, which uses the parallel compression system, performs best. It achieves a bandwidth of over 1500 Mbps for the bigger datasets under analysis. The benefit of the parallel compression pipeline system becomes more noticeable when comparing it to ‘Gipfeli Naive’, which uses compression but without the pipeline. In that case, the bandwidth is below 750 Mbps, and no compression presents better results (almost 1000 Mbps).

Figure 10 presents the results using a 100 Mbps network. As can be observed, except for ‘Lzo Naive’, all the approaches using compression obtain better results than not using compression. In these experiments, using the parallel compression pipeline system with Lzo provides a performance increase of over 100 Mbps compared to using compression without the pipeline. However, this is not the case for Gipfeli, where compression without the pipeline (‘Gipfeli Naive’) presents better results. In fact, ‘Gipfeli Naive’ obtains the best values in this scenario, attaining a maximum bandwidth of over 300 Mbps. We implement the parallel compression pipeline system to hide the compression and decompression time by overlapping it with the transfer time. Nevertheless, the compression system without a pipeline performed better when using Gipfeli in the last experiments. As explained in previous sections, the client of the compression system without the pipeline compresses all the data before sending them over the network. The server decompresses the data once received. Despite not using a pipeline to hide the compression and decompression times, this approach performs better in this specific case. The reason for this is explained next.

In Section 4.4, we evaluated the parallel compression pipeline system and selected the best configuration parameters depending on the network bandwidth and the compression library used. Table 3 presents a summary of that selection. According to that table, the resources needed (i.e., threads and data chunks) to optimally configure the parallel compression pipeline when using 100 Mbps and 10 Mbps networks are lower than the resources required for a 1 Gbps network. This means the bottleneck is in the network’s speed rather than the compression and decompression times. In Table 4, we show the sum of compression and decompression times and also the compression ratio when both Gipfeli and Lzo compress: (i) the 1 KB chunk used by the parallel compression pipeline (‘Gipfeli Pipeline’ and ‘Lzo Pipeline’), and (ii) the 8MB data compressed by the compression system without pipeline ‘Gipfeli Naive’ and ‘Lzo Naive’ (remember that, as shown in Table 3, 1 KB is the optimal chunk size for our parallel compression pipeline when using these libraries over 100 Mbps and 10 Mbps networks). As we can observe in Table 4, both the compression ratio and the time used to compress and decompress data increase with data size in both libraries. Notice that a compression ratio increment means that the transfer time is reduced. Thus, given that the bottleneck is the speed of the network bandwidth in slow networks such as 100 Mbps or 10 Mbps, having a larger compression ratio may compensate for the larger compression/decompression time. This is what happens with Gipfeli. For this reason, ‘Gipfeli Naive’ performs better than the ‘Gipfeli Pipeline’ in this specific case. This does not happen with ‘Lzo Naive’ because compression and decompression times take too long and do not compensate for the reduction in transmission time.

Finally, Figure 11 presents the results using a 10 Mbps network. As we can see, all the approaches using compression obtain better results than those that do not use compression. Additionally, compression without a pipeline performs better than the parallel one. ‘Gipfeli Naive’ obtains values near 30 Mbps with some peaks of 37 Mbps, while ‘Gipfeli Pipeline’ obtains values near 20 Mbps. ‘Lzo Pipeline’ obtains values near 22 Mbps with some peaks over 25 Mbps, while ‘Lzo Naive’ obtains values near 30 Mbps with some peaks over 35 Mbps. As noted before, these results are due to the fact that the compression ratio and the time used to compress and decompress increase proportionally to the data size. The size of the data to be compressed is even more relevant in this case because the 10 Mbps network is ten times slower than the 100 Mbps network.

### 4.6. To Send or Not to Send Compressed Data

As commented, the parallel pipeline compression system presented in this paper compresses the data on-the-fly before sending them over the network. However, depending on the exact data contents, the resulting compressed data size can be larger than the original uncompressed data. This happens more often for small messages. For that reason, in this section, we evaluate an improvement consisting in checking the size of the compressed data in order to decide whether to send the data compressed or uncompressed. Note that the latter will also avoid the time needed to decompress the data in the server.

We evaluate this improvement using the Gipfeli compression library with 1 Gbps, 100 Mbps, and 10 Mbps networks. Figure 12, Figure 13 and Figure 14 show the results obtained by ‘No Compression’, where no compression libraries are used, ‘Gipfeli Pipeline’, where the compressed chunk is always sent, and ‘Gipfeli Pipeline V2’, where the parallel compression pipeline decides whether sending the uncompressed chunks instead of the compressed ones. These results were normalized to the ‘No Compression’ ones. Thus, ‘No Compression’ values are always 1, while ‘Gipfeli Pipeline’ and ‘Gipfeli Pipeline V2’ obtain values greater than 1 when they improve ‘No Compression’.

Figure 12 illustrates the results obtained with a 1 Gbps network. As we can see, ‘Gipfeli Pipeline V2’ obtains better values than ‘Gipfeli Pipeline’ when using data sizes lower than 16 KB. Both obtain similar results with bigger data sizes, with ‘Gipfeli Pipeline’ being slightly better. Both approaches improve the bandwidth obtained by ‘No Compression’ when data sizes are greater than 256 KB.

Similar results are obtained using a 100 Mbps network, as shown in Figure 13. Again, ‘Gipfeli Pipeline V2’ only achieves better values than ‘Gipfeli Pipeline’ for data sizes lower than 16 KB. Both enhance ‘No Compression’ for data sizes greater than 32 KB. Note that the 16 KB threshold is lower than the one shown in Figure 12 for a 1 Gbps network, which was 256 KB.

Last but not least, the experiment shown in Figure 14 presents the results obtained using a 10 Mbps network. The differences between ‘Gipfeli Pipeline V2’ and ‘Gipfeli Pipeline’ become lower as we reduce the network bandwidth. The network is the bottleneck, so they obtain similar results in this case. With data sizes lower than 16 B, ‘Gipfeli Pipeline V2’ achieves better performance than ‘Gipfeli Pipeline’. For values over 16 B, there are a lot of variabilities. In this scenario, ‘Gipfeli Pipeline V2’ and ‘Gipfeli Pipeline’ improve ‘No Compression’ for data sizes larger than 4 KB. It should be noted that this threshold from which the pipelined versions improve the approach of not using compression reduces together with the network’s speed. Thus, compression becomes more relevant as the network is slower, as was expected.

To better understand the previous results, Figure 15 shows the percentage of compressed chunks used by ‘Gipfeli Pipeline V2’ when using data with different sizes. A 0 value means no compressed information has been sent because the compressed result is larger than the original uncompressed data. On the other hand, a 100 value indicates all data sent over the network have been compressed before being sent. As can be observed, the system does not compress the data transferred over the network until the data size is equal to or larger than 32 B. From that value, the compression is relevant. For some data sizes, such as 2 KB, we can detect lower percentages of compressed chunks. In general, the percentage of chunks that are sent compressed is larger for larger data sizes.

In this section, we evaluated an improvement to our parallel compression pipeline system, which consists of checking the compressed data size to decide whether to send the compressed or uncompressed data. As observed in the experiments, this improvement only provides better results for smaller data sizes. For larger data sizes, it performs similarly to the previous version of the system. In addition, as network speed decreases, the achieved improvement is smaller.

## 5. Conclusions

In this paper, we have presented the implementation and performance evaluation of a parallel compression pipeline intended to be used within the communication layer of remote GPU virtualization solutions such as rCUDA. The purpose of this parallel compression pipeline is to artificially increase network bandwidth by reducing the size of data being exchanged between the CUDA application and the remote server that owns the real GPU. Compression is carried out on-the-fly so that applications are not aware that data exchanged with the remote GPU server are compressed. Remote GPU virtualization solutions are transparent to CUDA applications. Therefore, the source code of applications does not need to be modified in order to benefit from using a remote GPU.

The performance analysis presented in this paper has been carried out for different network bandwidths. Namely, we have considered 1 Gbps, 100 Mbps, and 10 Mbps networks. We have also used several data patterns, given that compression greatly depends on the exact data to be processed. In this regard, in the paper, we have presented results using the data exchanged to/from the GPU during the execution of TensorFlow deep learning applications.

The analysis presented in the paper raises several interesting conclusions. The first one is that, by using compression, network bandwidth can be artificially increased up to a factor of 2× when the size of data exchanged with the remote GPU is large enough. In conrast, for very small data sizes, using compression reduces network performance. However, we have seen that as networks becomes slower, the benefits of using compression become much more noticeable. Despite improving the ‘No Compression’ results, pipeline compression works worse than naive compression, where all the data are compressed before sending them, with slow networks and traced data. That occurs because the transfer time is the bottleneck in this scenario, so compressing all the data obtains a better compression ratio and, therefore, a better transfer time than pipeline compression.

There are several opportunities that this work provides for future work. For instance, given that the remote server owns a GPU, using the remote accelerator for compressing and decompressing data in the server should provide further performance improvements, thus making the use of on-the-fly compression even more appealing. Another possibility for future work is using adaptive compression; that is, instead of using a single compression library, several of them could be simultaneously considered depending on the size of the data to be compressed as well as the actual network bandwidth and CPU cycles available (CUDA programs are concurrent applications that concurrently exchange data with the GPU from their threads). Considering the use of different compression libraries depending on the data type being exchanged is also possible. To that end, a simple neural model could be used to quickly decide which compression algorithm to use. Moreover, a communication layer that does not make use of compression for very small data sizes should also be considered. As can be seen, the study in this paper opens a lot of future research directions to further improve available network bandwidth.

Finally, in this paper, we have focused on improving network bandwidth. However, this research is intended to be used in the edge scenario, so future work should also consider a thorough energy analysis.

## Figures and Tables

**Figure 1 sensors-24-04649-f001:**
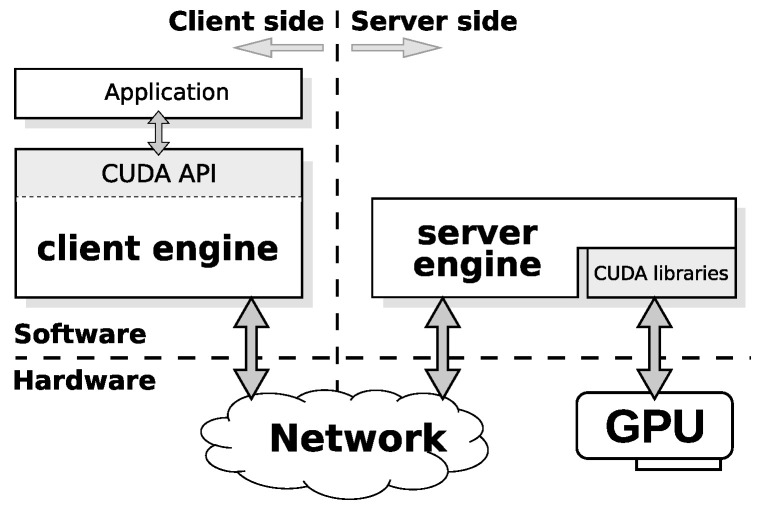
General architecture of remote GPU virtualization solutions.

**Figure 2 sensors-24-04649-f002:**
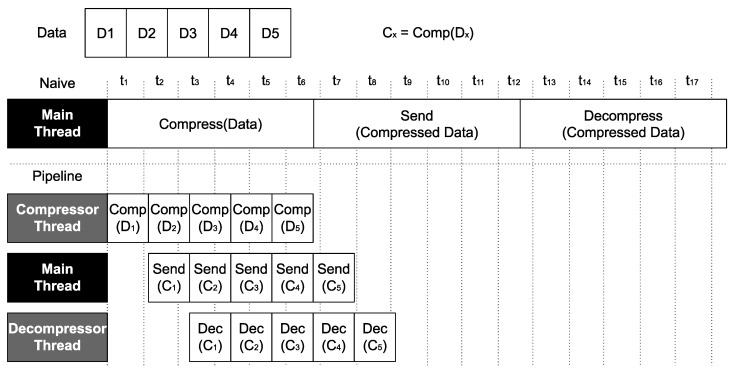
Comparison between a naive compression approach and the parallel compression pipeline proposed in this paper.

**Figure 3 sensors-24-04649-f003:**
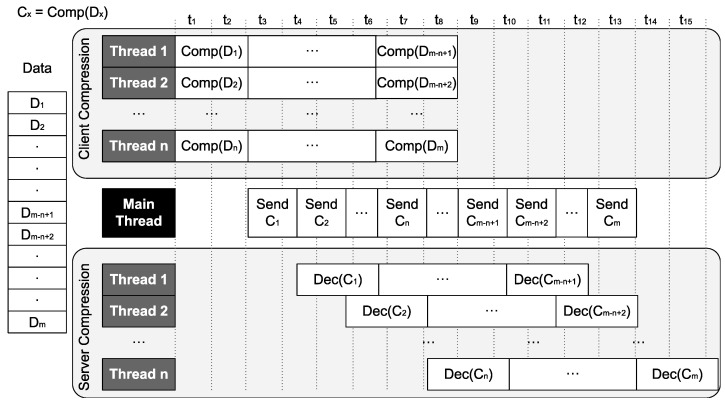
High-level diagram of the implementation of the parallel compression pipeline using *n* compression/decompression threads and *m* data chunks.

**Figure 4 sensors-24-04649-f004:**
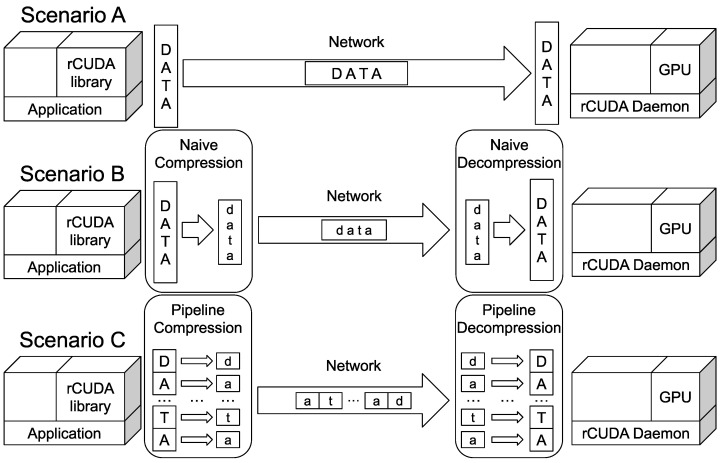
Scenarios explored in the experiments.

**Figure 5 sensors-24-04649-f005:**
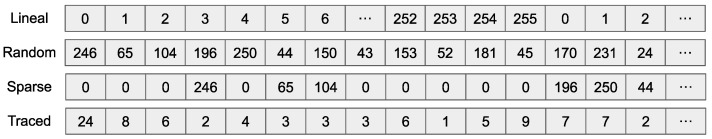
Data used in experiments.

**Figure 6 sensors-24-04649-f006:**
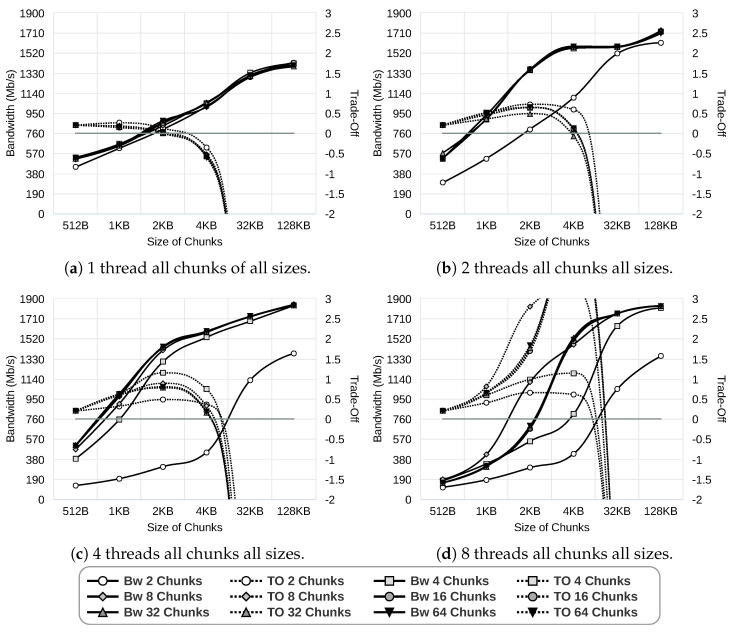
Study of data chunk sizes using 1 Gbps network and Gipfeli compression library. The horizontal line in the plots corresponds to a trade-off equal to zero.

**Figure 7 sensors-24-04649-f007:**
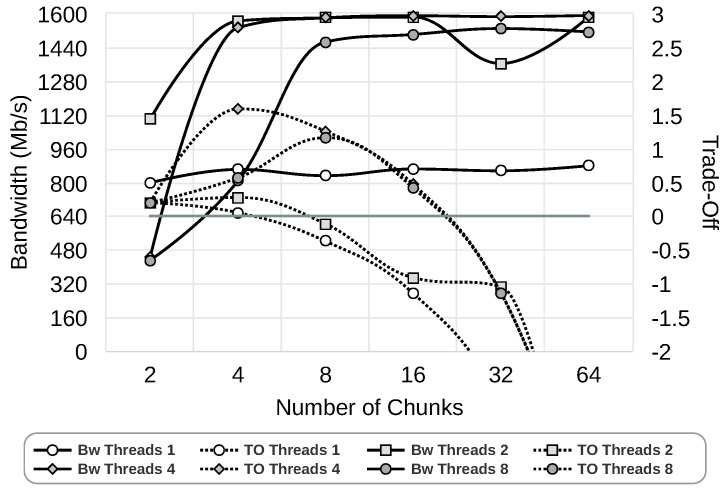
Study of the number of data chunks using 1 Gbps network and Gipfeli compression library.

**Figure 8 sensors-24-04649-f008:**
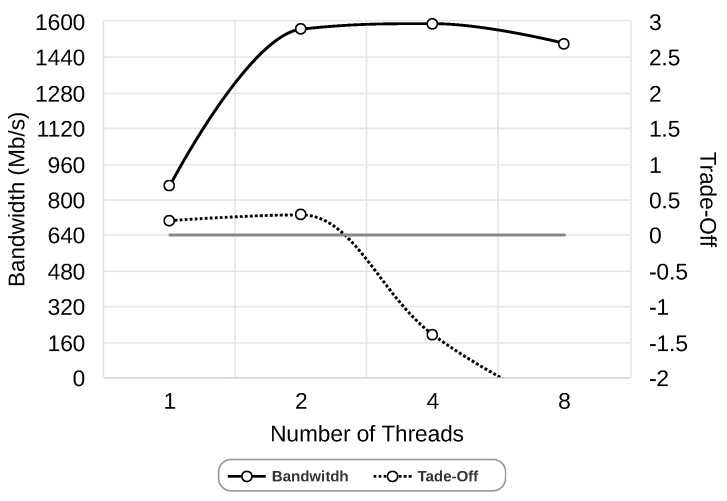
Study of the number of threads using 1 Gbps network and Gipfeli compression library.

**Figure 9 sensors-24-04649-f009:**
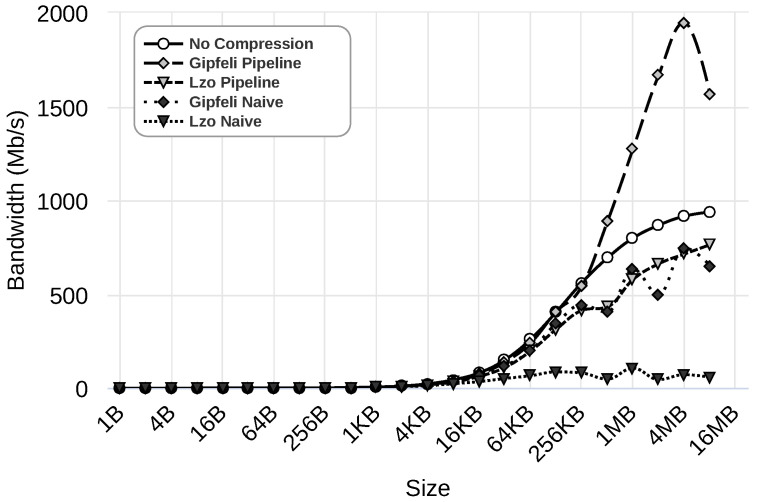
Results using a 1 Gbps network. Different scenarios are considered: ‘No Compression’ (rCUDA without using compression); ‘Gipfeli Pipeline’ and ‘Lzo Pipeline’ (rCUDA using the parallel compression pipeline system with Gipfeli and Lzo, respectively); ‘Gipfeli Naive’ and ‘Lzo Naive’ rCUDA using compression with Gipfeli and Lzo, respectively, without the pipeline. The scenarios are evaluated with datasets of different sizes. More details about these scenarios can be found in Section 4.1.

**Figure 10 sensors-24-04649-f010:**
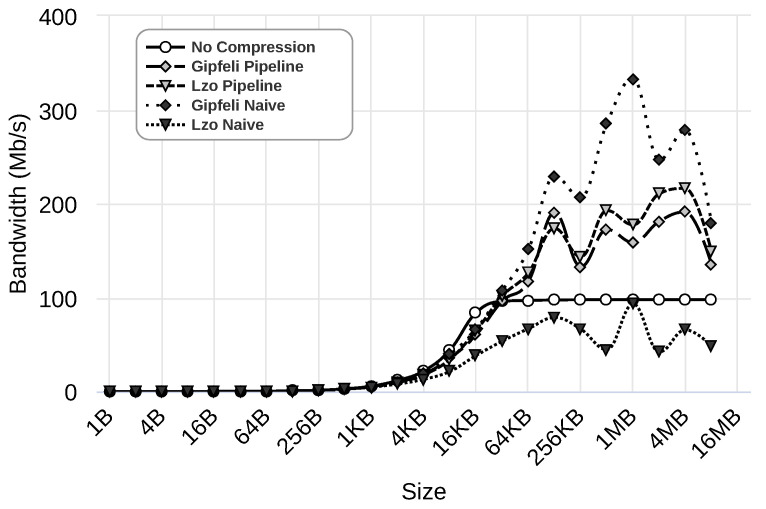
Results using a 100 Mbps network. Different scenarios are considered: ‘No Compression’ (rCUDA without using compression); ‘Gipfeli Pipeline’ and ‘Lzo Pipeline’ (rCUDA using the parallel compression pipeline system with Gipfeli and Lzo, respectively); ‘Gipfeli Naive’ and ‘Lzo Naive’ rCUDA using compression with Gipfeli and Lzo, respectively, without the pipeline. The scenarios are evaluated with datasets of different sizes. More details about these scenarios can be found in Section 4.1.

**Figure 11 sensors-24-04649-f011:**
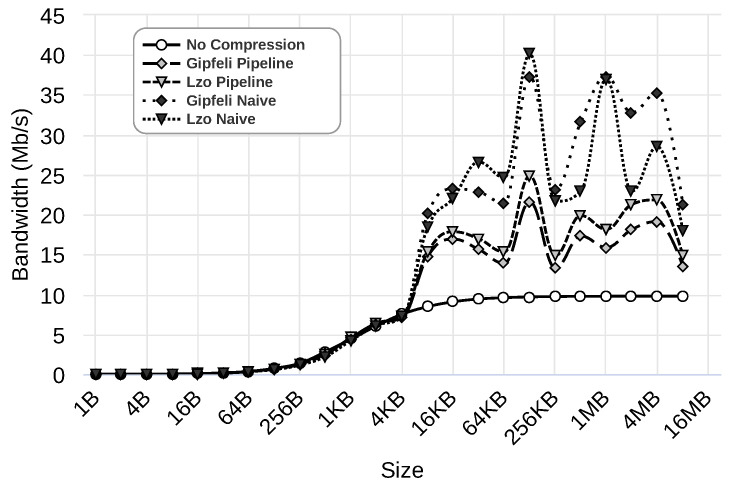
Results using a 10 Mbps network. Different scenarios are considered: ‘No Compression’ (rCUDA without using compression); ‘Gipfeli Pipeline’ and ‘Lzo Pipeline’ (rCUDA using the parallel compression pipeline system with Gipfeli and Lzo, respectively); ‘Gipfeli Naive’ and ‘Lzo Naive’ rCUDA using compression with Gipfeli and Lzo, respectively, without the pipeline. The scenarios are evaluated with datasets of different sizes. More details about these scenarios can be found in Section 4.1.

**Figure 12 sensors-24-04649-f012:**
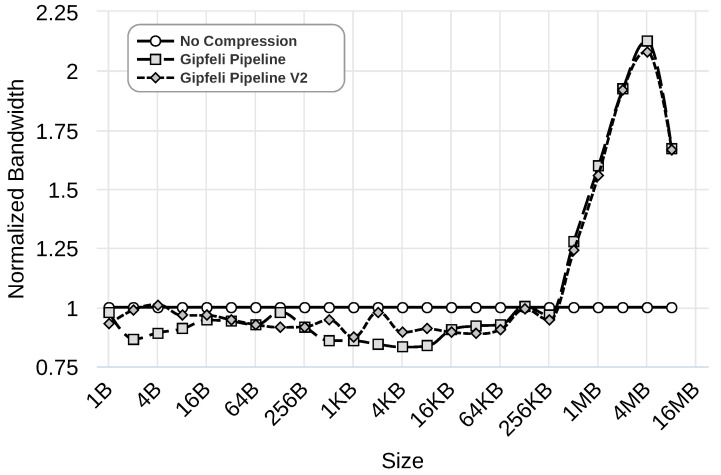
Normalized bandwidth obtained over a 1 Gbps network by ‘Gipfeli Pipeline’ and ‘Gipfeli Pipeline V2’. The latter only sends the compressed data if the size is smaller than the original uncompressed data.

**Figure 13 sensors-24-04649-f013:**
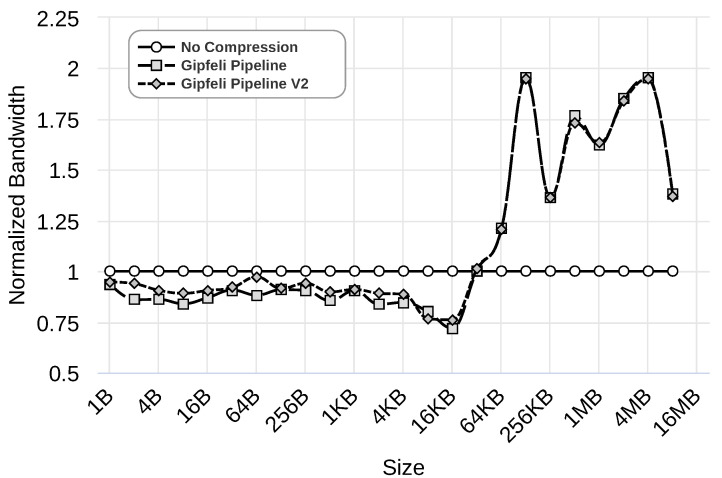
Normalized bandwidth obtained over a 100 Mbps network by ‘Gipfeli Pipeline’ and ‘Gipfeli Pipeline V2’. The latter only sends the compressed data if the size is smaller than the original uncompressed data.

**Figure 14 sensors-24-04649-f014:**
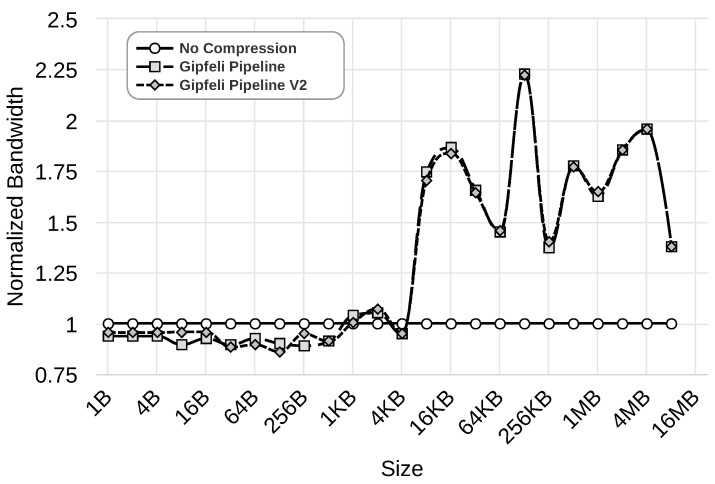
Normalized bandwidth obtained over a 10 Mbps network by ‘Gipfeli Pipeline’ and ‘Gipfeli Pipeline V2’. The latter only sends the compressed data if the size is smaller than the original uncompressed data.

**Figure 15 sensors-24-04649-f015:**
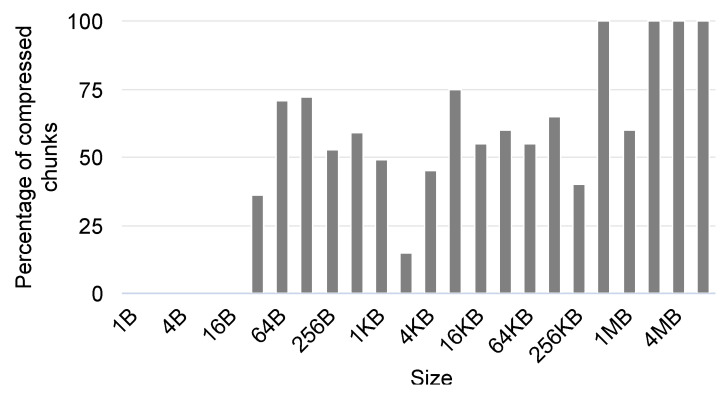
Percentage of compressed chunks sent to the network with ‘Gipfeli Pipeline V2’ when using data with different sizes.

**Table 1 sensors-24-04649-t001:** Parameters used to analyze the performance of the parallel compression pipeline.

Number of Threads	Number of Chunks	Size of Chunks
1	2	512 B
2	4	1 KB
4	8	2 KB
8	16	4 KB
	32	32 KB
	64	128 KB

**Table 2 sensors-24-04649-t002:** Summary of the best parameters for the parallel compression pipeline system using Gipfeli and 1 Gbps network for each amount of threads used in the pipeline.

Number of Threads	Number of Chunks	Size of Chunks
1	4	2 KB
2	4	4 KB
4	16	4 KB
8	16	4 KB

**Table 3 sensors-24-04649-t003:** Summary of the best parameters for the parallel compression pipeline system using different compression libraries and networks.

Network	Library	Threads	Chunks	Size (KB)	Ratio	Compression + Decompression Time (μs)	Bandwidth (Mb/s)
1 Gbps	Snappy	1	2	4	2.25	17.81	1345.68
	Gipfeli	2	4	4	2.36	55.39	**1565.04**
	Lz4	4	8	2	2.24	74.26	1190.93
	Lzo	8	8	2	2.52	159.54	763.29
100 Mbps	Snappy	1	4	1	1.49	7.96	127.41
	Gipfeli	1	4	1	1.54	21.84	135.02
	Lz4	1	4	1	1.73	40.43	134.24
	Lzo	2	4	1	1.94	91.56	**148.42**
10 Mbps	Snappy	1	4	1	1.49	7.96	12.74
	Gipfeli	1	4	1	1.54	21.84	13.51
	Lz4	1	2	1	1.73	40.43	13.48
	Lzo	1	2	1	1.94	91.56	**14.87**

**Table 4 sensors-24-04649-t004:** Compression ratio and the total compression and decompression time obtained using Gipfeli and Lzo when compressing 8 MB traced data and a 1 KB chunk of this data.

Library	Data	Compression + Decompression Time (μs)	Compression Ratio
Gipfeli	Chunk (1 KB)	21.84	1.54
	Whole (8 MB)	40,468.28	2.22
Lzo	Chunk (1 KB)	91.56	1.94
	Whole (8 MB)	1,662,723.10	2.65

## Data Availability

The data presented in this study are available on request from the corresponding author. The data are not publicly available due to privacy restrictions.

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
