# Peer review of "A Parallel Compression Pipeline for Improving GPU Virtualization Data Transfers"

_sensors, 2024, doi:10.3390/s24144649_

Round 1

Reviewer 1 Report

Comments and Suggestions for Authors

This paper proposes an approach which is very similar to the fog computing. It aims to train the computation-intensive AI workload on edge devices but the edge devices are constrained by the limited computation power on the localhost. Thus it is necessary to leverage the computation power provided by remote GPUs. However, data transfer can be huge and the network in turn will become the new bottleneck. The proposed solution is simple: try to do some compression before transferring the data to remote GPUs.  Following this path, the authors developed a pipeline scheme to conduct the compression. The experiments also demonstrate the effectiveness of the approach.

In general, the design makes sense and the topic is worthwhile in practical use. However, I think there is less novelty in the paper since many such works have emerged even before the era of AI. In fog computing and edge computing field, the cloudlet paradigm has been proposed and many follow-up works have been done. 

Despite the story is old, the authors have conducted comprehensive evaluation in the paper and proposes many insights. Besides, the writing is very clear and easy to follow. As a result, I still feel this paper is well above the bar of Sensors and vote for acceptance. 

Author Response

Comment 1: In general, the design makes sense and the topic is worthwhile in practical use. However, I think there is less novelty in the paper since many such works have emerged even before the era of AI. In fog computing and edge computing field, the cloudlet paradigm has been proposed and many follow-up works have been done.

Despite the story is old, the authors have conducted comprehensive evaluation in the paper and proposes many insights. Besides, the writing is very clear and easy to follow. As a result, I still feel this paper is well above the bar of Sensors and vote for acceptance.

Response 1: Thank you for your comments.

Reviewer 2 Report

Comments and Suggestions for Authors

The authors present how to implement a parallel compression pipeline that can be used to improve GPU virtualization data transfers. Just as the authors stated in the introduction section, this paper is not about proposing novel compression algorithms, but is about the parallel implementation of the existing algorithms. Very detailed experimental results are conducted and reported, and the network bandwidth can be increased by a factor of up to 2. However, there are still two main concerns that need to be addressed. 

(1) In fact, there has similar research about the parallel implementation, for example, in section 2.2, Peterson and Reiher's work was a parallel version and could hide the compression and decompression time behind the transfer time.  It is a little confused for readers to understand what is the difference between your parallel implementation idea and the existing idea.

(2) In the experiment section, it lacks comparisons with existing works, and it is strongly recommended that performance comparisons between different parallel implementations should be carried out to show the advantage of the presented method over the existing method. 

(3) It provides an efficient implementation of the parallel compression pipeline to increase the network bandwidth in the GPU virtualization applications.

(4) The authors focus on evaluating different pipeline configurations and finding the best parameters for the pipeline. However, some necessary comparisons between the presented method and the existing method are suggested to add, for example, is the presented method having advantage over other methods with the same parameter configurations?

(5) The experiments have been carried out in different parameter configurations for different network bandwidths. And they show the network bandwidth can be increased up to a factor of 2. 

Author Response

Comment 1: In fact, there has similar research about the parallel implementation, for example, in section 2.2, Peterson and Reiher's work was a parallel version and could hide the compression and decompression time behind the transfer time. It is a little confused for readers to understand what is the difference between your parallel implementation idea and the existing idea. 

Response 1: Peterson and Reihera developed an adaptive compression and evaluated it by comparing it with existing solutions. Then, they use this adaptative library in a parallel version. This last point is similar to our results but with a significant difference. Their solution splits data into chunks of 32KB, independently of data or network type. Our study is focused on finding the best parameters (threads, number of chunks, and size of chunks ) to create a parallel compression pipeline. Note that these parameters vary depending on the data, compression library, and network used. Our study seeks to guide the creation of parallel compression pipeline algorithms given an existing compression library.

We extended Section 2.2 of the paper to clarify this point.

Comment 2: In the experiment section, it lacks comparisons with existing works, and it is strongly recommended that performance comparisons between different parallel implementations should be carried out to show the advantage of the presented method over the existing method.

Response 2: Thank you for your valuable feedback. We appreciate your suggestion to include comparisons with existing works.

While there is related research in the field, after an exhaustive search, we have not found other studies evaluating the same aspects as our work.

Our primary objective is to develop a robust methodology for creating parallel pipeline compression libraries. This focus on methodology is intended to provide a foundation for further research and development in this area.

Comment 3: It provides an efficient implementation of the parallel compression pipeline to increase the network bandwidth in the GPU virtualization applications.

Response 3: As the reviewer points out, our study is focused on improving the width of the network bandwidth in GPU virtualization systems.

Comment 4: The authors focus on evaluating different pipeline configurations and finding the best parameters for the pipeline. However, some necessary comparisons between the presented method and the existing method are suggested to add, for example, is the presented method having advantage over other methods with the same parameter configurations?

Response 4: As mentioned previously, there are not enough studies that evaluate a parallel pipeline compression to improve network communications, and the few existing ones do not share the necessary libraries or sources to perform direct comparisons.

Our primary goal is to develop a comprehensive methodology for creating a parallel library and to optimize pipeline configurations and parameters within this framework. Establishing this foundation is crucial for future research and practical applications.

Comment 5: The experiments have been carried out in different parameter configurations for different network bandwidths. And they show the network bandwidth can be increased up to a factor of 2.

Response 5: That is right. Thanks to the study, we managed to improve the bandwidth up to 2x. Thanks for the feedback.

Reviewer 3 Report

Comments and Suggestions for Authors

The paper aims at improving GPU Virtualization data transfers, it present the design and mplementation of a parallel pipelined compression system to be used in remote GPU virtualization solutions.

This paper does not propose any novel compression algorithms,just make use of well-known compression algorithms to implement and evaluate our parallel pipelined proposal.

Comprehensive experiments and evaluations are conducted to perform a thorough performance analysis to investigate how the different parameters of the pipeline influence the overall bandwidth. 

This work is easy to follow and understandable, however, there are some major revisions should be considered.

1. This paper's work presented in the introduction is relatively limited, the key innovation points of the article can be summarized  in the introduction.

2. The paper conducted extensive and thorough experiments to validate the proposed approach.But the experimental comparison methods were based on open-source websites, and the contrast algorithms were from 2021 and 2022, which are considered outdated and lack academic rigor.

3. There are quite some mistakes and presentation errors throughout the paper. The authors should pay attention to the formats of formulas. 

Above all, the author should correct and revise the manuscript carefully.

Comments on the Quality of English Language

There are quite some mistakes and presentation errors throughout the paper. The authors should pay attention to the formats of formulas. 

Author Response

Comment 1: This paper's work presented in the introduction is relatively limited, the key innovation points of the article can be summarized in the introduction.

Response 1:  We have tried to better present the contributions of our work in the introduction.

Comment 2: The paper conducted extensive and thorough experiments to validate the proposed approach. But the experimental comparison methods were based on open-source websites, and the contrast algorithms were from 2021 and 2022, which are considered outdated and lack academic rigor.

Response 2: That's right, the compression libraries are based on open-source code, but we do not think they are outdated. Gipfeli and Lzo could be considered older compression libraries. However, LZ4 and Snappy are well-known compression libraries whose repositories are updated weekly. There are many compression libraries available, and Smash offers 41 different ones. We chose these libraries because of their differences, not just their modernity.

Our goal is to develop a mechanism for a parallel compression pipeline using a given compression library rather than to compare and select the best existing library. By including a range of libraries, both old and new, we aim to widen the design space. This approach ensures that our study remains robust and relevant, even as new compression libraries emerge that might use different principles or approaches.

Despite their age, these libraries remain cited in research papers, indicating they maintain enough academic rigor to be included in this study. This diversity in selection enhances the validity of our conclusions and prepares our framework for future developments in compression technology.

We have reinforced the use of these compression libraries in Section 4.2 of the paper.

Comment 3: There are quite some mistakes and presentation errors throughout the paper. The authors should pay attention to the formats of formulas.

Response 3: Thanks for the comments.

We have reviewed the paper exhaustively.

Round 2

Reviewer 2 Report

Comments and Suggestions for Authors

1. The authors have added necessary explanations to give the major differences between the present work and the existing works, which can make readers more clear about the aim of the paper. 

2. The objective is to develop the method for creating parallel pipeline compression libraries, and the network bandwidth can be increased up to 2x. 

3. The manuscript has been improved, and the quality of presentation is good.